# Tunable macroscale structural superlubricity in two-layer graphene via strain engineering

Charalampos Androulidakis [1,4], Emmanuel N. Koukaras [1,2,4], George Paterakis [1,3], George Trakakis [1] & Costas Galiotis [1,3 ✉]

Achieving structural superlubricity in graphitic samples of macroscale size is particularly challenging due to difficulties in sliding large contact areas of commensurate stacking domains. Here, we show the presence of macroscale structural superlubricity between two randomly stacked graphene layers produced by both mechanical exfoliation and chemical vapour deposition. By measuring the shifts of Raman peaks under strain we estimate the values of frictional interlayer shear stress (ILSS) in the superlubricity regime (mm scale) under ambient conditions. The random incommensurate stacking, the presence of wrinkles and the mismatch in the lattice constant between two graphene layers induced by the tensile strain differential are considered responsible for the facile shearing at the macroscale. Furthermore, molecular dynamic simulations show that the stick-slip behaviour does not hold for incommensurate chiral shearing directions for which the ILSS decreases substantially, supporting the experimental observations. Our results pave the way for overcoming several limitations in achieving macroscale superlubricity using graphene.

[1] Institute of Chemical Engineering Sciences, Foundation of Research and Technology-Hellas (FORTH/ICE-HT), Stadiou Street, Platani, Patras 26504, Greece. [2] Laboratory of Quantum and Computational Chemistry, Department of Chemistry, Aristotle University of Thessaloniki, GR-54124 Thessaloniki, Greece. [3] Department of Chemical Engineering, University of Patras, Patras 26504, Greece. [4] These authors contributed equally: Charalampos Androulidakis, Emmanuel N. Koukaras. ✉email: c.galiotis@iceht.forth.gr

Two-dimensional (2D) layered solids such as graphite comprise single layers held together via van der Waals forces and possess extraordinary mechanical properties[1]. This weak interlayer coupling significantly affects the properties of multi-layers. From the mechanics point of view, the in-plane tensile fracture strength tends to decrease with the increase in thickness[2], and recent experiments have also shown a decrease in the in-plane compressive strength as a result of premature cohesive shear failure[3]. Experiments performed using friction force microscopy (FFM) by shearing an atomic force microscopy (AFM) tip over the surface of 2D crystals (graphene and hBN) with various numbers of layers in thickness, have indicated that 2D materials possess thickness-dependent friction properties; for graphene, the friction has been found to increase with the decrease in thickness[4]. Due to its significance for the use of graphitic materials as lubricants in a number of applications, the friction behaviour of graphene[5–11] and graphite[12–15] has been the subject of extensive research.

Graphite is a well-known solid lubricant, a property that originates from the low interlayer shear strength between individual graphene layers[12,16]. Structural superlubricity can occur between two solid surfaces of crystalline nature when they are stacked in an incommensurate configuration. This is also manifested by the mismatch of their lattice constants, as for example in the case of graphene and hexagonal boron nitride[17]. In fact, there are a number of factors that affect the shear behaviour of graphite and consequently the friction, such as the dimensions of the test specimen and the shearing direction of the graphene layers in respect to each other[16]. For graphite flakes with large dimensions (>10 μm), the interlayer shear strength tends to increase, and the lubricant behaviour does not hold because of the presence of many commensurately stacked domains in the large contact areas that cause mechanical interlocking between the individual layers[16]. Recently, it was reported that superlubricity can be achieved at the micron scale when hexagonal boron nitride is sheared over graphite; these 2D crystals have an intrinsic lattice constant mismatch that favours sliding for all directions[18]. Lubricant behaviour has also been observed for single-layer graphene, even at the macroscale, when sliding a surface coated with graphene against another surface with diamond like carbon and nanodiamond particles, which makes graphene a very versatile, thin and transparent coating material for use in a variety of applications[19]. To date, the experimental approach for studying interlayer shear behaviour involves sliding an AFM tip over the sample. To our knowledge, measurements with large contact areas and macroscale observation of superlubric behaviour in graphene has not been done as yet, and has been accomplished only for carbon nanotubes (CNTs)[20]. Another issue that plays a crucial role in the lubricant properties of graphene and graphite is the presence of water. Recent studies show that high relative humidity enhances the lubricant behaviour of graphene on $SiO_2$[21], and also water can be intercalated between two graphene layers and affects the interlayer interactions[22].

Herein, we report direct measurements of the interlayer shear stress in incommensurately stacked bilayer graphenes produced both by mechanical exfoliation of graphite and chemical vapour deposition (CVD) synthesis, simply supported on a polymer substrate. As each specimen consists of two randomly overlapping monolayers, its 2D Raman peak is a single peak indicating AA[23] stacking in contrast to the composition of four subpeaks of a Bernal-stacked (AB) bilayer[24]. The specimens are subjected to tensile strain on flexed beams under a Raman microscope as explained previously[25,26]. The top single layer is selected to be smaller than the bottom layer in such a way that it is not in contact with the polymer, and therefore is stretched only by the strain transferred solely by the bottom layer. In fact, the different levels of strain in the two layers lead to 2D peak splitting that allows the monitoring of strain applied on each layer and the estimation of interlayer shear stress using continuum theory. On the contrary, the 2D peak of an AB-stacked bilayer shifts as one unit under strain[27]. As argued herein, the measured range of shear stresses (which are relaxed/reduced during sliding) indicates a superlubric behaviour. This behaviour persists macroscopically since the CVD information is collected from an extensive contact area in the range of $mm^2$, which is much larger than in any previously reported works. Molecular dynamic (MD) simulations are also performed to further elucidate the experimental findings. Specifically, we examine the effect of wrinkles and shearing direction on the ISS of bilayer graphene. The wrinkles tend to decrease the ISS between the graphene layers, while shearing in incommensurate chiral directions breaks the stick-slip behaviour accompanied with a dramatic decrease in ISS.

## Results and Discussion

**Preparation and testing of an exfoliated sample**. Graphene flakes were prepared by mechanical cleavage of graphite using the scotch tape method and deposited directly on a polymer substrate (PMMA/SU-8). The two-layer graphene was formed by mechanically folding a single layer during deposition. In Fig. 1a, b, an optical image and the corresponding Raman spectra of the single-layer graphene along with its folded part are presented. As revealed by the Raman spectra recorded under the same conditions, the intensity of the 2D peak of the folded single layer[28,29] is approximately four times higher than that expected from a single layer due to the changes induced to the double-resonance process compared with a single layer. Additional differences are also recorded/identified, such as the decrease in the full width at half maximum (FWHM) and the shift to a higher-frequency value for the sample at rest (Fig. 1b). The underlying physics of this phenomenon has been discussed in detail elsewhere[29].

By bending the polymer substrate with a four-point-bending apparatus, the bottom single layer is subjected to tension as it is in contact with the polymer, while the top layer of the folded part is strained solely by the bottom graphene layer (see Fig. 1c), allowing to capture the shearing mechanism in a graphene/graphene interface. In Fig. 1d, a schematic of the experimental setup is shown with two single-layer graphenes stacked in an incommensurate state. The figure also depicts the formation of Moiré patterns as discussed in detail below. Mappings of the Raman shifts were performed along a line of the specimen under tensile strain, which is several microns in length and fully spans the folded bilayer and in part the single layer. The evolution of the 2D spectra of the folded bilayer for various levels of tensile strain is shown in Fig. 2a. At zero strain, the peaks are symmetric or have a slight asymmetry, depending on the mismatch in the level of residual strain of each individual layer (of the bilayer). The 2D peak fitted very well with two Lorentzian functions for all strain levels. Increasing the applied tensile strain causes the 2D peak to split to two subpeaks that eventually become fully distinct from each other due to the different level of actual strain in each individual graphene layer (Fig. 2a). The clear peak splitting allows the detection of wavenumber shift per increment of strain for each layer in the folded region that can be directly compared to the corresponding shift of the stand-alone graphene. The strain transfer results on the two graphene layers are presented in Fig. 2b, c. The changes observed in the lineshape and frequency of the 2D peak, as well as the strain transferred, which is clearly demonstrated through the Raman shift, show that the two graphene layers are in contact with each other. The measured values for the bottom layer in the folded region were on average $\sim-43.7\,cm^{-1}\,\%^{-1}$ that compares well with the value of $\sim-48.7\,cm^{-1}\,\%^{-1}$ obtained from the single-layer area

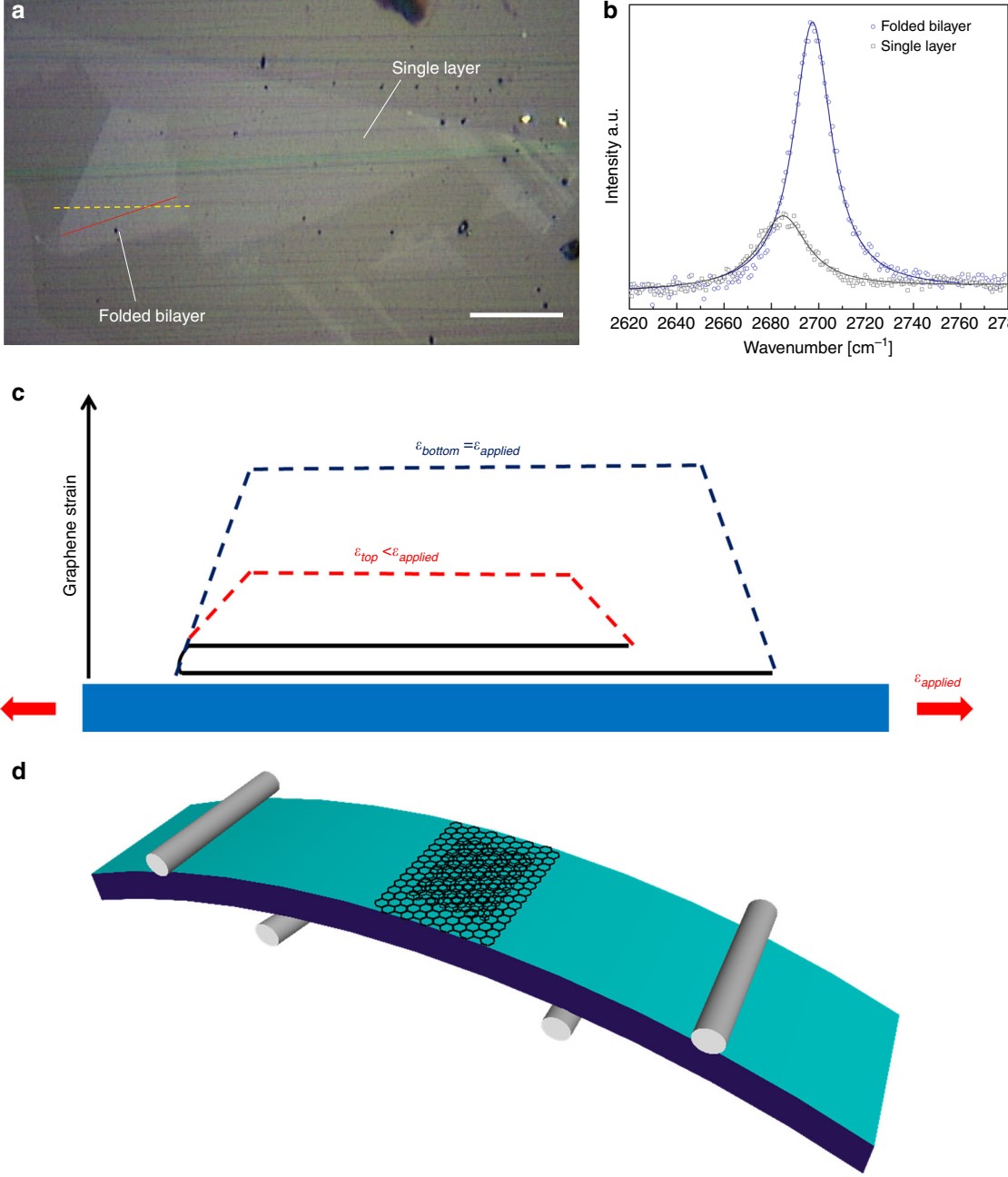

**Fig. 1 Sample characterization and experimental setup. a** Optical image of the single layer with a folded part at the left. The dashed yellow line indicates the scan line, and the red line marks the presence of a wrinkle (see the AFM image in Supplementary Fig. 1). The scale bar is 20 microns. **b** Spectra of the 2D peak of the single and folded areas. **c** Schematic representation of the stress transfer mechanism from the polymer to the bottom layer and from the bottom layer to the top. **d** Schematic of the experimental setup is shown with two single-layer graphenes stacked in an incommensurate state.

as expected[30]. The small difference noted in the shift rate for the bottom graphene is because the average shift obtained from the folded area is somewhat reduced by the build-up from the edges (see sketch of Fig. 1c). Accounting only for the points near the central area where the maximum redshift occurs, a similar shift rate is obtained for the two locations. On the other hand, however, the shift of the 2D peak for the top layer of the folded bilayer graphene is about half that of the bottom layer, measured at $\sim -23.1\ cm^{-1}\ \%^{-1}$ (Fig. 2b). Furthermore, slipping is observed at applied strain levels as low as 0.2%, indicating premature interlayer failure (denoted with black circles in Fig. 2b). The position of the 2D peak drops abruptly to lower wavenumbers from one strain

level to the next, suggesting that slipping between the graphene and the polymer occurs after this point.

In Fig. 3, Raman maps across the length of both single layers that form the folded bilayer are presented. The shape of the stress-transfer curve from the polymer to the inclusion for the bottom layer (Fig. 3a) is, as expected, governed by polymer–graphene shearing that leads to stress build-up from the flake edges and the attainment of a plateau at the middle of the flake[25]. This mechanism is a result of the strain transfer with friction, which leads to linear strain profiles at the edges with constant interlayer frictional stress and the length required for strain-transfer increases with the increase in the applied load[31]. As discussed

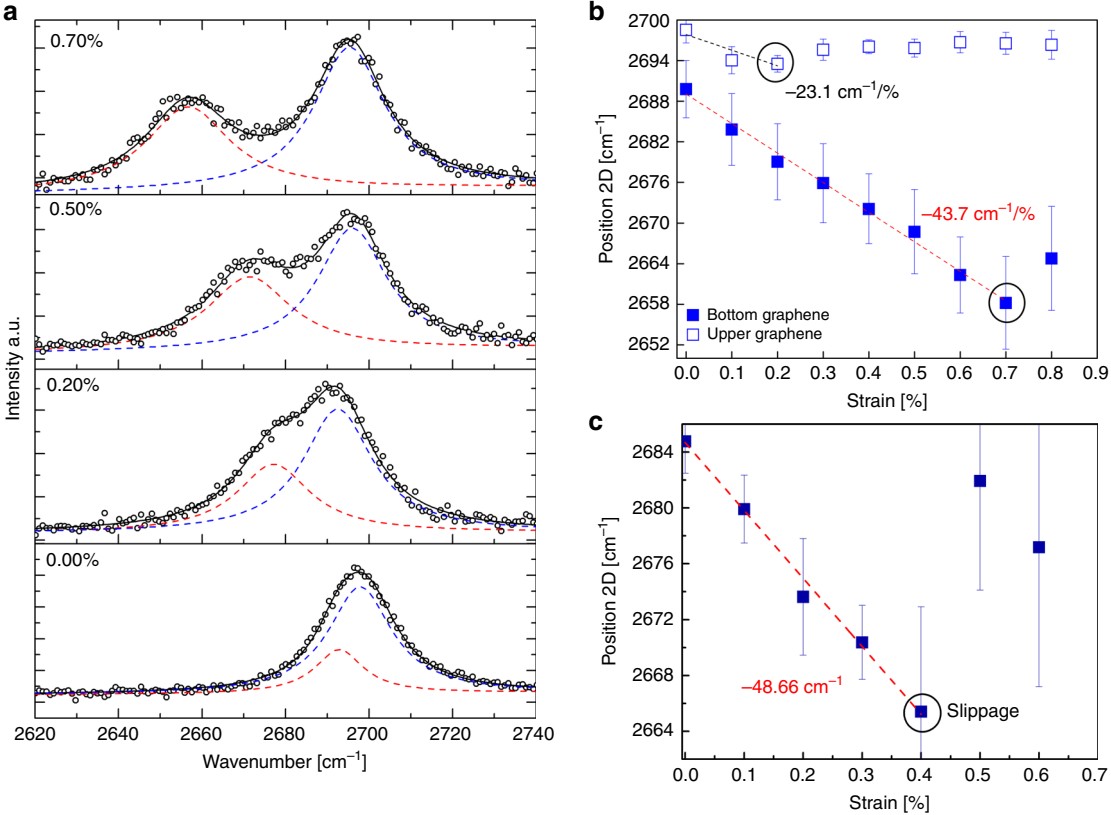

**Fig. 2 The shift of the 2D peak under tension. a** The evolution of the 2D peak for the folded bilayer graphene under various levels of tensile strain. **b** The shift of the 2D peak of the bottom and top single-layer graphenes of the folded bilayer. **c** The shift of the single-layer graphene solely. The error bars represent standard deviation.

below, this is a crucial point that has not received attention to date, and holds for the case of a graphene–graphene interface. To avoid any confusion, we refer to the shear between graphene/polymer as interfacial shear stress (IFSS) and between graphene/graphene as interlayer shear stress (ILSS). We observe that the strain profile of the top graphene layer follows a similar pattern, as the predominant mechanism here is also shearing, but in this case between the two individual graphene layers. The edge on the right (as shown in Fig. 1 and recorded in the plot of Fig. 3b) is clearly representative for measuring the ILSS since the strain is transferred purely by shearing from the bottom layer. At the left edge (where the fold is present), there is also an axial tensile force that stretches the top layer. This is depicted in Fig. 3b, where the strain build-up deviates from that expected for pure shearing, and the value of strain is not constant. Based on the strain build-up and the balance of forces in the graphene/graphene interface, the developed interlayer shear stress can be measured as presented in the schematic of Fig. 1c. It is noted that at a distance of ~10 microns from the outer (left) edge towards the inner part of the flake (see Fig. 1a red line and by AFM in Supplementary Fig. 1), there is a wrinkle that disrupts the strain transfer and acts like a line on the edge. One crucial point is also derived from these results: the length required in order to reach the maximum ILSS in a graphene/graphene is 6 microns, and accounting for both edges, 12 microns are required. This length explains the differences and supports the results for superlubric behaviour of graphite[16] as discussed in detail later.

As mentioned above, the top single layer is stretched only by the strain transferred through shearing from the bottom graphene. By balancing the shear to axial forces at the graphene/graphene interface, the interlayer shear stress can be estimated—as also in

the case of graphene/polymer stress transfer—from the following equation[25]:

$$\left(\frac{\partial \varepsilon}{\partial x}\right)_{T \equiv 298K} = -\frac{\tau_t}{n t_g E} \Leftrightarrow \tau_t = -n t_g E \left(\frac{\partial \varepsilon}{\partial x}\right)_{T \equiv 298K} \quad (1)$$

where $\varepsilon$ is the strain, $\tau_t$ is the interfacial shear stress, $E$ is the Young's modulus of graphene, $n$ is the number of layers of the graphene and $t_g$ is the thickness of a single-layer graphene. The interfacial shear stress per increment of strain can be obtained by employing Eq. (1) having extracted the $\partial \varepsilon / \partial x$ slopes from the Raman measurements. For the interface graphene/polymer, the maximum IFSS is ~0.45 MPa, in agreement with previous results[25,32]. The maximum ILSS of the graphene/graphene interface is estimated to be ~0.13 MPa, which is ~3–4 times lower than the IFSS based on the above analysis. This value is in very good agreement with the results obtained elsewhere[33]. Furthermore, as shown in Fig. 4 in the case of graphene/polymer interface, an IFSS plateau is formed above 0.2% strain, which indicates that the interface survives strains at least up to 0.8%. In contrast, the results for the graphene–graphene system clearly show slipping beyond 0.2% strain, and a sudden drop of the ILSS to zero value. Essentially, this means that up to that strain level, the material system is in the regime of superlubricity, and for further tension (≥0.2%), sliding occurs, which enhances this behaviour. In contrast, such an effect is not observed in commensurable (Bernal stacked) bilayers up to quite high tensile strains[26]. It is worth remarking on the strain profiles for the top graphene layer shown in Fig. 4. For strains >0.50%, the ILSS fluctuates along low values in the range of ~0.04–0.05 MPa, which indicates that the flake is practically sliding from that level of

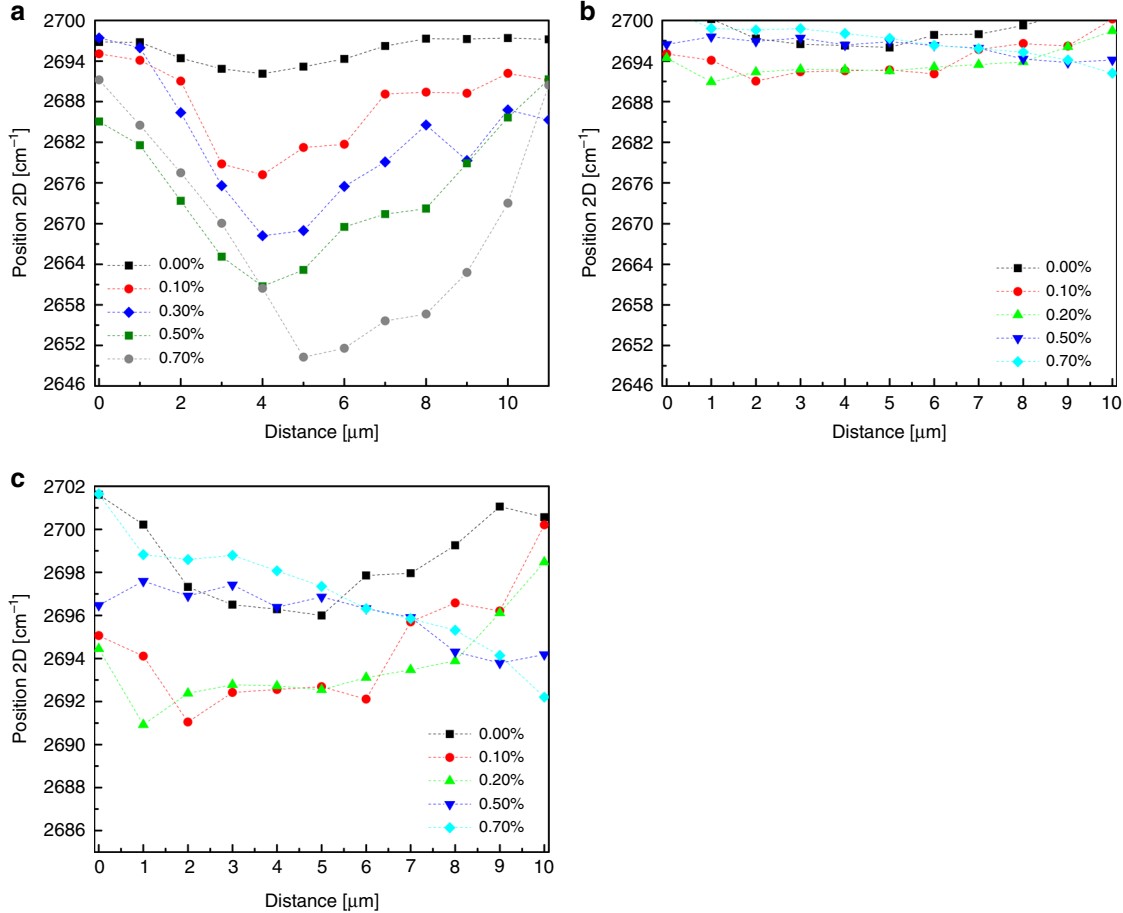

**Fig. 3 Strain-transfer mechanism.** Maps of the frequency of the 2D peak for various levels of strain showing the distribution profile of the frequency of the 2D Raman peak across the length (**a**) of the bottom and (**b**) top single-layer graphene of the folded bilayer. In (**a**), (**b**), the data are plotted with the same scale on the y axis for comparison, and in (**c**), a zoom version of the results of (**b**) is presented for clarity.

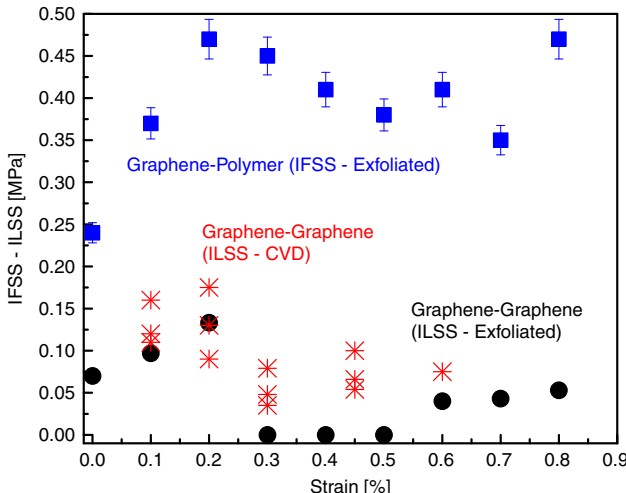

**Fig. 4 Interface and interlayer shear stress.** The interfacial shear stress (IFSS) for the cases of exfoliated single-layer graphene/polymer, exfoliated graphene/graphene and CVD graphene/graphene (ILSS) for various levels of tensile strain. The error bars represent standard deviation.

strain onwards[16], as a result of the axial tensile force acting at the right edge of the flake. Another very interesting point is that the maximum and minimum ILSS for the two-layer graphene is also in very good agreement with recent results for Bernal

bilayer graphene, shearing graphite mesas and with a hBN/graphene heterojunction[18]. The measured ILSS in the range of ~0.07–0.13 MPa, as well as the 0.04–0.05 MPa values recorded for the higher strain levels, are in the corresponding range of frictional stresses associated with superlubric behaviour[16]. The present results demonstrate microscale superlubric behaviour of a two-layer exfoliated graphene, in agreement with the results obtained from turbostratic graphite mesas[16].

**Preparation and testing of the CVD sample.** In practical applications, CVD graphene is mainly used, which can be produced at any shape and size (even roll-to-roll)[34]. Thus, it remains to be established whether such behaviour is also observed for CVD graphene sheets, and more importantly if the superlubric sliding can be applied in macroscale. As mentioned above, we have repeated a similar experiment by sequential stacking of CVD single layers with dimensions of a few mm in length. Care was taken in order for the top single layer to be of smaller size (in both dimensions) than the bottom graphene layer that is in contact with the polymer (see Supplementary Fig. 3). A PMMA fragment that is detected on the top of the assembly was not removed to avoid introducing defects to the system. Details for the preparation of the sample are provided in the Methods section. Using this bilayer structure, we are able to capture the stress build-up from the top-layer edge and measure the interlayer shear stresses similarly to the case of the folded exfoliated flake.

The shift rate of the bottom CVD single layer is $\sim -14.4$ cm$^{-1}$ %$^{-1}$ that agrees well with the results obtained from other studies[35,36], and

the average shift of the top layer is $\sim-5.9\,cm^{-1}\,\%^{-1}$. It is remarkable that the ratio of shift rates between the top and bottom layer is similar for both exfoliated and CVD graphene, with values of 0.52 and 0.37–0.45 for the exfoliated and CVD, respectively. This finding indicates that in both cases, the axial strain transferred to the top layer from the bottom graphene layer for the case of non-Bernal stacking is about half of the corresponding strain induced to the bottom layer from the polymer substrate. In Fig. 5, the Raman maps for the top CVD layer are presented for various levels of strain. For such large sheets within the micron scale, the strain build-up from the sheet edges that occurs over much smaller distances is difficult to be observed. CVD graphene contains structural defects, such as wrinkles with amplitude of a few nm, that reduce the stress-transfer efficiency and therefore the magnitude of the transmitted strain[36]. In Fig. 5d, the topography of the bottom-deposited CVD layer is presented by AFM scanning.

The polycrystalline nature of the CVD graphene leads to randomly stacked areas of large extent between the stacked CVD graphene sheets. As seen in Fig. 5, with a scan step of 1 micron, we observe stress build-up in some areas (red dashed line in Fig. 5), while in other areas, the distribution forms a plateau or just fluctuates. This is a consequence of (a) the polycrystalline nature of CVD graphene that results in random stacking, and also (b) the presence of wrinkles that affects the stress-transfer efficiency. The wrinkles give rise to discontinuities in the strain transfer, creating small areas with local strain build-ups similar to the islands in the case of CVD on polymer[37]. The maximum ILSS obtained for the CVD/CVD from the edge of the bilayer is in the range of ~0.04–0.16 MPa, in very good agreement with the results obtained from the exfoliated bilayer (Fig. 4), and in the range of frictional stress for superlubric behaviour. We must note that this value corresponds to the areas that a build-up is observed, and the ILSS is much smaller for areas for which the strain profile is a plateau, as a consequence of the random stacking or by the presence of wrinkles. Moving towards the inner part of the bilayer, the strain profile is not smooth, but as is observed from Fig. 5a, local stress build-ups occur over a length of a few microns, with slopes that have similar values within the above-mentioned range of shear stress. As seen in Fig. 4, a drop in the ILSS is observed at ~0.30% due to the slip from the edges, which begins to take place. In the strain regime 0.30–0.60%, the superlubric behaviour is more pronounced as evident by the values of ILSS. For the strain level of 0.60%, a local stress build-up is observed, but this time due to compressive strain at the edge of the top graphene (initial length in Fig. 5a). The graphene starts to slip from the edge, and the graphene is compressed. This phenomenon matches very well with the results obtained from simulations for both the slipping and the strain level[38].

Having established the stress-transfer mechanism between single-layer graphenes with length of a few microns, we further examined a CVD/CVD interface over a large (~3 mm) distance. Due to the amount of time required to perform such Raman maps, we selected a tensile strain of 0.50% to perform extended maps in order to examine if a similar behaviour is reproduced at the macroscale. In Fig. 5c, the results of the mm-scale Raman mapping are presented. The extensive scan confirms that the behaviour we observe for a 30-micron scan close to the CVD graphene edge is reproducible for a length of 3 mm. The frequency of the 2D peaks coincides with the values at the edge, showing that periodically the same strain distribution across the mm scale occurs. Thus, this behaviour holds for the whole two-layer CVD graphene. The experiment is fully presented in Supplementary Note 2. This last result, that confirms superlubric behaviour at the macroscale, constitutes the central and most important finding of this work, and signifies that conditions and

mechanisms that inhibit manifestation of superlubricity for macroscale graphitic specimens can be overcome between even, as few as, two polycrystalline graphene layers. The mechanism leading to this behaviour is analyzed and discussed below in detail.

The present experimental approach is vastly different than the usually adopted method of dragging a layer of graphite[12], or shearing an AFM tip of a few nanometers in diameter over graphene[4]. Controllable uniform straining without size limitations can be applied, limited only by the time taken for collecting the Raman data. Moreover, the Raman maps collect information from large contact areas. The different level of strain in each single layer induces a mismatch between the lattice constant (the bottom layer is under higher strain than the top, and thus the lattice constant is somewhat different under increased tension), and thus, gradual incommensurable stacking occurs, leading to interfacial sliding as interlayer shear strength is overcome. The effect of lattice mismatch induced by strain has been examined by simulations that show robust superlubric behaviour when sliding a graphene on strained graphene[39,40]. As is experimentally evident, our approach provides an alternative for achieving macroscale superlubricity using two CVD graphene layers.

We note here that two lattices in incommensurate state form Moiré patterns that affect the spatial distribution of strain, and consequently the interlayer shearing depends on the Moiré characteristics[41–43]. Despite the stress concentration that might be present in such cases, the mismatch in the lattice constant between the two layers eliminates such effects. For example, as explained in the case of hBN/graphene interface with inherent lattice constant mismatch[18], the possible presence of small friction anisotropy does not alter the overall system behaviour. Moreover, MD simulations on shearing a graphene layer on a strained graphene show that the friction dramatically decreases with the increase in graphene size. This is because the large contact areas result in a much larger length than that of Moiré patterns, and the friction force tends to the value of the incommensurate state[39]. Thus, such effects can hardly affect the interlayer shear stress measured at the micron/mm scale of our experiments.

Besides the lattice constant mismatch that was discussed above, the presence of wrinkles that exist at grain boundaries of CVD graphene and the relative shearing direction are critical factors for the manifestation of superlubricity that need to be examined to understand their effect on shearing of the two graphene layers[16]. In Fig. 5d, AFM scans of the bottom graphene layer are presented, showing the wrinkled (or rough structure due to the underlying polymer) structure formed during deposition and transfer of the CVD graphene, which evidently affect the stress-transfer mechanism. Previous studies showed that when an AFM tip is sheared over wrinkles, it causes an increase in friction[44,45]. This might lead one to expect that the presence of wrinkles in the graphene–graphene interface could increase the interlayer friction and consequently the ILSS; however, this is not in accordance with the present findings. The case of stress transfer of wrinkled graphene on polymer is different, and we refer the reader to previous work[26,46,47]. Moreover, under tension, the wrinkles are flattened out, and new wrinkles are formed laterally to the applied tension[48,49]. The second factor is the strong dependence of the shear strength to the relative direction of shearing of the graphene sheets[12]. It is thus of great importance to examine these two effects by simulations on shearing of a graphene layer over another one. We performed molecular dynamic (MD) simulations to examine the influence of the presence of wrinkles on the stress-transfer efficiency in bilayer graphene, as well as to access any dependence of the interlayer shear stresses on chirality of shear directions.

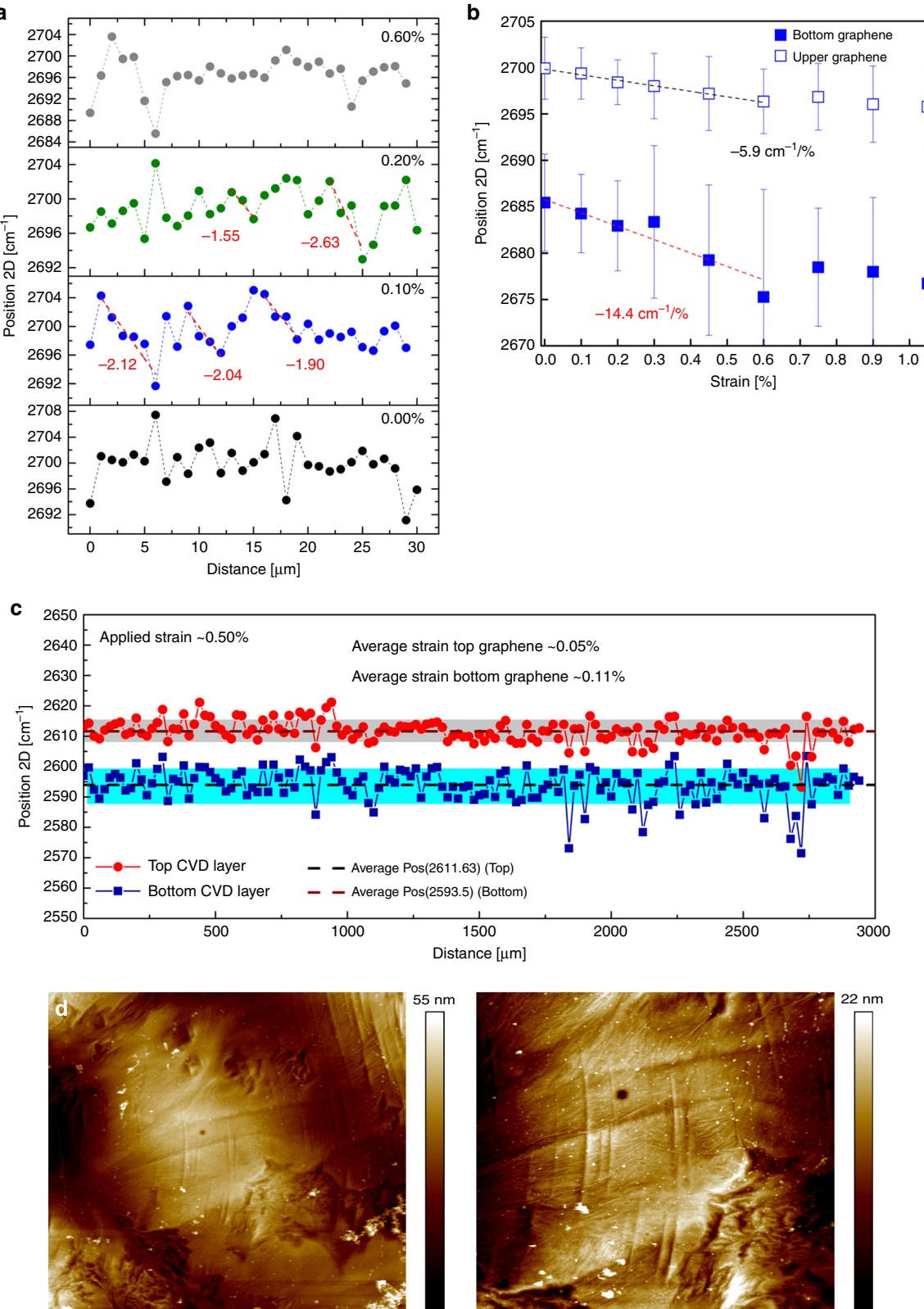

**Fig. 5 Shearing of CVD graphene–graphene sample. a** The position of the 2D peak versus distance for the top CVD single-layer graphene under various levels of tension. In-plot values correspond to slopes shown with dashed lines. Similar values of the slopes indicate similar ILSS. **b** The average shift of the 2D peak per increment of strain for the bottom and top CVD single layers for a distance of 30 microns. **c** Raman mapping over a 3-mm distance of a CVD/CVD bilayer specimen with a scan step of 10 μm. The average position is presented by the dashed lines and the standard deviation with the shaded colours. The corresponding average values of strain are also mentioned in the graph. Note: The experiment in (**c**) performed with a laserline excitation of 785 nm resulting in different frequencies compared to (**b**), for which a laserline of 514 nm was used. In SI, the response for the whole strain regime is presented. **d** AFM image (left) of the as-deposited CVD on the polymer bar, and (right) its x2 magnification. The scale bar is 2 microns.

**Molecular dynamic (MD) simulations**. To illustrate the effect of wrinkles on the interlayer shearing between two graphene layers, molecular dynamics simulations were performed on supported bilayer graphene that had been previously subjected to compressive strain of –0.6% in order to induce the formation of wrinkles. The results show that the presence of wrinkles, either in the parallel or perpendicular direction, does not contribute to any increase in the ILSS. On the contrary, wrinkles or out-of-plane deformations due to the roughness of the substrate contribute to the lowering of the ILSS by a factor of ~2, if other phenomena do not come into play. The overall analysis is presented in Supplementary Note 3 along with a discussion.

We now move on to the investigation of the effect of the shearing direction on the interlayer shearing that is the most crucial factor. There is a large range regarding the reported values of interlayer shear strength of graphite[12,16,32,50,51]. This range has been recently significantly narrowed down, and the crucial role of the shearing direction was identified[16]. By adopting a technique that was originally proposed for self-cleaning of graphitic surfaces[52], Liu et al. performed shearing experiments on graphitic mesas that were capped with $SiO_2$[16]. In doing so, they managed to estimate narrow shear strength value ranges for two cases: when the sheared flake was in the lock-in state, and when it was in the superlubric-incommensurate state. From their analysis on the self-retraction force, they reported an upper-bound estimate for the superlubric shear strength of $\tau^{upper}_f = 0.02–0.04$ MPa for flakes of ~10 microns in length, and from their analysis on the deformation of a tungsten tip, they reported a value for the lock-in shear strength of $\tau_{lock-in} = 0.1 \pm 0.04$ GPa. The superlubric shear strength upper bound corresponds to relatively small graphite flakes, and has a strong size and direction dependence[16]. Here, we examined the effect on the ILSS of the relative orientation of the two graphene layers. The top layer was moved along the bottom layer armchair direction (to be clear, the notation used here for direction is the same as that used for length or growth direction of graphene nanoribbons (see Methods section further below). To account for the effect of stacking orientation, distinct simulations were performed with the top layer having been constructed with five different chiral angles, namely, zigzag (0°), 7.5°, 15°, 22.5° and armchair (30°). The ILSS plots that emerge are given in Fig. 6. The slip-stick pattern that corresponds to armchair-over-armchair case stands out. Even though the aim here is to qualitatively capture the effect, nevertheless the maximum ILSS values obtained, of ~65–80 MPa, are very close and well within the range of the lock-in shear strength of Liu et al., of $\tau_{lock-in} = 0.1 \pm 0.04$ GPa. The effect is also captured by repeating the simulations employing the AIREBO potential[53], as discussed in Supplementary Note 3[18,54–57].

For all of the intermediate (to the main) chiral directions, a significant drop in ILSS is immediately apparent. This ILSS reduction is also encountered when the simulations are repeated at a temperature of 1 K, so this effect does not emerge from thermal rippling. The computed ISS values for the turbostatic stacking reach as low as ~1 MPa that is higher than the reported upper bound of superlubric shear strength mentioned above, but in very good agreement with the results from the same work obtained with a tip that exhibits plastic deformation[16]. A reduction is also possible to occur from the presence of wrinkles as discussed here and in Supplementary Note 3, but it is expected that other effects also come into play and are discussed in what follows. Another factor that could affect the interlayer interactions is the presence of relative humidity, which could be a source for the observed discrepancies regarding graphene superlubricity between simulations and experiments[22]. In the work by Liu et al.[16], the results were obtained from graphite mesas with thickness of hundreds of nm. It was recently shown that the interlayer shear strength of few-layer graphene is thickness dependent, and tends to decrease with the increase in the thickness[3].

Another crucial factor is the size of the examined samples. The strain transfer shearing mechanism is remarkably similar in qualitative terms to that of graphene–polymer, and thus, a certain length is required in order to have efficient strain transfer as evident by the experiments. From the results of both exfoliated and CVD experiments, this length is estimated to be maximum ~4–12 microns (accounting both edges, see Fig. 3b) deduced from the results of both exfoliated and CVD graphenes. This length is in excellent agreement with the results obtained by Liu et al.[16] where the self-retraction phenomenon begins to break down. Moreover, the ILSS obtained from bilayer graphene blister has a maximum value of 0.06 MPa and an average of 0.04 MPa[58], which is in good agreement with the lower values of the present work obtained for exfoliated graphene and somewhat lower from the average values. A small reduction in the value of ILSS is to be expected when the graphene size is smaller than the transfer length, which is a plausible explanation for this small difference and in agreement with the results by Liu et al.[16]. At any rate, the CVD bilayer manifests macroscale superlubricity.

In summary, we examined the interlayer shearing behaviour of a bilayer graphene with random stacking. The shearing mechanism revealed that in order to have fully strain transfer between two graphene layers, a length of 12 microns is required in order to reach the maximum ILSS. Further, we examine a bilayer consisting of CVD graphene layers of cm dimensions. The random stacking breaks the continuous shear mechanism and the ILSS is orders of magnitude lower than the shearing at chiral directions, leading to the creation of local periodic strain build-ups. The tensile strain also induces a lattice mismatch, and along with the random stacking, leads to macroscale superlubricity. In practical applications, two contacted surfaces can be coated with a single layer of graphene preferably with a small residual tension, which can lead to a substantial decrease in frictional stresses as demonstrated experimentally.

## Methods

**Sample preparation and mechanical testing**. Highly ordered pyrolytic graphite (HOPG) was mechanically cleaved using a scotch tape and the graphitic materials deposited on a PMMA-SU-8 substrate. The SU-8 photoresist was spin-coated on the top of a PMMA bar of thickness ~3 mm with rotational speed of ~4000 rpm. The single-layer graphene and its folded part were identified by the lineshape of the 2D Raman peak. A four-point-bending jig under the Raman microscope was used for simultaneously recording Raman spectra and mechanically loading the sample. Laser lines of 785 nm and 514 nm were used for the execution of the experiments. The strain was applied incrementally with a step of ~0.1–0.15% for all cases.

**Preparation of the CVD graphene–graphene sample**. For the fabrication of large-area CVD two-layer graphene, the following procedure was followed. The bottom layer needs to be on the top surface of the polymer in order to deposit the second layer on its top in direct contact (Supplementary Fig. 3). In the CVD sample consisting of graphene–copper (the graphene on the other side of the copper has already been removed), PMMA was spin-coated over the graphene at ~1000 rpm for about ~30 s (resulting in thickness of the PMMA ~170 nm) creating a sample PMMA–graphene–copper. The PMMA employed was dissolved in anisole solution of 3% concentration. Having this sample ready for deposition on the polymer bar, a thin layer of PMMA was spin-coated on the polymer bar with a speed of ~3000 rpm for 3 s, followed by immediate attachment with the PMMA–graphene–copper. The attachment was between the two PMMA layers. Thus, the relatively soft PMMA layer is attached to the PMMA–graphene–copper. The sample is then left under low pressure for a few hours. Attaching the PMMA layers during their soft phase allows their robust attachment. We note that if the PMMA is not freshly spin-coated on polymer bar, the two layers do not attach well to each other. The copper was then removed by exposing the sample to ammonium persulfate [0.1 M], leaving on the top a CVD graphene. A second single-layer graphene was deposited on the top of the bottom layer using the usually adopted approach of wet transfer of CVD using a PMMA layer as support[59]. The graphenes were rinsed with distilled water four to five times[59] in order to clean their surfaces. Extra caution was taken in order that the top layer is supported only by the bottom graphene. This was succeeded by

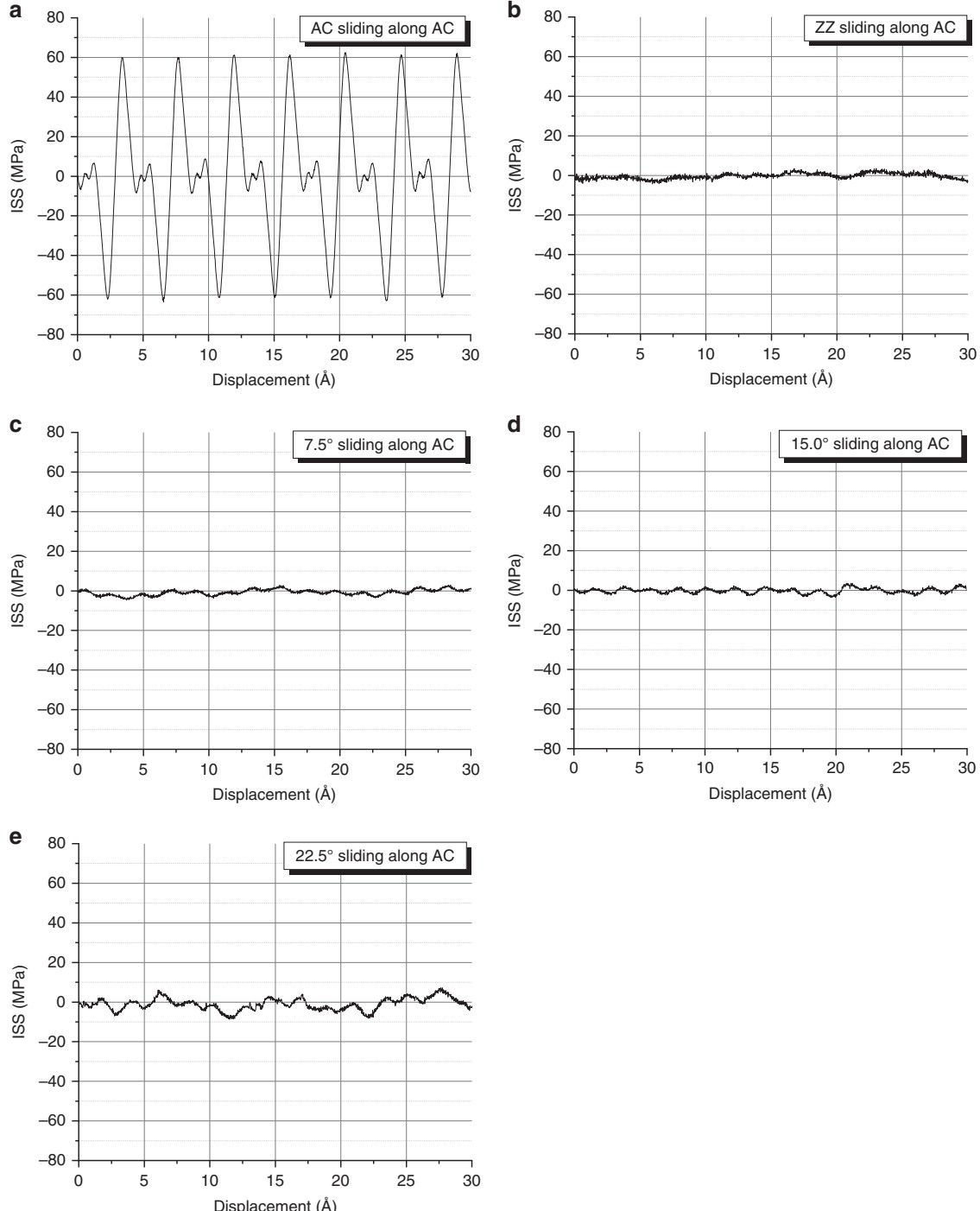

**Fig. 6 Interlayer shear stress from MD simulations.** ILSS values for various chiral directions at a temperature of 300 K, specifically, **a** armchair, **b** zigzag, **c** 7.5°, **d** 15.0° and **e** 22.5°, sliding with respect to the armchair sublayer. It is observed that when shearing a monolayer graphene in achiral directions relative to the underline graphene, the stick-slip motion breaks down and the ILSS is significantly decreased. This is performed by employing the LCBOP potential.

depositing a relatively large CVD to the bottom (i.e. ~2 × 1 cm for length and width, respectively), and for the top layer about half the dimensions of the bottom layer. As mentioned below, the initial CVD sample has a dimension of 7 cm² × 7 cm²; thus, no size limitations were encountered during this procedure. Finally, the sample was left to dry under nitrogen flow, and zero relative humidity over 24 h for the removal of any water molecules. We note that we did not subject the sample to heat to avoid the potential compressive strain induced by heating. A schematic of the procedure is given in the Supplementary Information (Supplementary Fig. 2).

**CVD graphene production**. Graphene was synthesized on copper foils by chemical vapour deposition (CVD) in an AIXTRON® BM Pro CVD chamber. Copper was

supplied by Viohalco® and used as the catalyst substrate. For the production, the foil was cut into 7 cm² × 7 cm², cleaned by isopropanol to remove any organic contamination and introduced into the CVD chamber. After the closure of the chamber, it was immediately pumped down to 0.1 mbar, and then a mixture of argon/hydrogen gases was introduced (250 sccm/50 sccm) under 25 mbar. The foil was heated at 1000 °C and was kept there for 5 min for annealing. Afterwards, the sample was cooled down to 925 °C, while methane was introduced into the chamber (10 sccm) as carbon feedstock to initiate the graphene growth on copper foil surface. After 5 min, the H₂ flow was terminated, the chamber was cooled down to 650 °C, CH₄ flow was terminated and finally the chamber was cooled down to room temperature under Ar atmosphere.

**Molecular dynamic (MD) simulations**. Molecular dynamics simulations were performed by employing the LCBOP potential[60] that offers Morse-type long-range interactions that exclude the nearest neighbours, and offers a suitably para-meterized short-range term that does not lead to unrealistic structural defects[60]. The simulations were fully dynamic for the dynamic particles of the system (as opposed to quasi-static simulations that employ additional algorithmic relaxation schemes[42]). Periodic boundary conditions were used in all cases, throughout. The bottom layer is periodic in both directions, was corrugated though compression as detailed in the Supporting Information and remained rigid during the sliding stage. The top layer was periodic in the direction normal to its displacement over the corrugated rigid bottom layer. The components of the forces along the displace-ment direction acting on all of the top-layer atoms were summed and averaged every 2000 time steps. A small time step of 0.5 fs was used. We denote here the simulations as "chiral" to assess the effect of chiral shear direction on ILSS, see main text. For the "chiral" set of simulations, computational cells of different sizes were used (provided in Supplementary Note 3), each adapted to conform to the sheet size constraints imposed by a given chiral angle. The naming convention followed here is analogous to that followed in the literature for chiral graphene nanoribbons, i.e. the naming is determined by the chirality of the edge along the shearing direction ($y$ axis) in analogy to nanoribbons that take their name (by convention) from the chirality of the long edge (length). With an AC bottom layer, top layers with chiral angles of 7.589°, 15.295°, 23.413°, AC and ZZ were exam-ined (chiral angles are with respect to zig-zag direction, thus relative angles of chirality between layers are given by $30° - \varphi$). The bilayer sheets lay over an interacting (Lennard–Jones) mathematical surface (wall) with parameter values $\varepsilon = 6.8$ meV and $\sigma = 3.133$ Å. All simulations were performed using the LAMMPS package[61]. Computational cells in the Supporting Information were visualized using OVITO[62].

## Data availability

The data that support the findings of the present study are available within the paper and its Supplementary file. Other data are available from the corresponding authors upon request.

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

## Acknowledgements

C.G. acknowledges the support from "Graphene Core 2, GA: 696656 which is implemented under the EU-Horizon 2020 Research & Innovation Actions (RIA) and is financially supported by EC-financed parts of the Graphene Flagship. C.A. and C.G. acknowledge the support from "APACHE", Active & intelligent Packaging materials and display cases as a tool for preventive conservation of Cultural HEritage", which is implemented under the EU-Horizon 2020. G.P. receives a scholarship from the General Secretariat for Research and Technology (GSRT) and the Hellenic Foundation for Research and Innovation (HFRI). G.T. and C.G. acknowledge the Bilateral German-Greek Research and Innovation Cooperation (CAERUS), implemented by the General Secretariat for Research and Technology (GSRT). E.N.K. acknowledges receiving funding for this project from the Hellenic Foundation for Research and Innovation (HFRI) and the General Secretariat for Research and Technology (GSRT), under grant agreement No. 1536.

## Author contributions

C.G. and C.A. designed the experiments. C.A., G.P. and G.T. prepared and characterized the samples, and C.A. performed the tensile experiments. E.N.K. performed the MD simulations. C.G. supervised the project. C.A., E.N.K. and C.G. wrote the paper with input from all authors.

## Competing interests

The authors declare no competing interests.
