## [Peer Review File · Nature Communications]

Reviewers' comments:

Reviewer #1 (Remarks to the Author):

The paper describes a novel method to observe frictional properties between graphene sheets. The method is based upon Raman imaging, where the strain in the film leads to shifts of the Raman peaks. Since the graphene sheets are deposited on a flexible beam, the strain can be continuously changed. Interestingly, the authors observe that the top graphene layer shows only very small strain values, which means that the top layer starts to slide above a certain threshold. Calculations of the effective shear stress show low values of the order of 0.1MPa, which are in agreement with other methods. Therefore, I propose to publish this interesting paper with minor changes.

A few minor points to be considered:

- The strain distribution shows maxima towards the edges. It would be interesting to explain this observation in more detail. Is it related to influence of the enhanced friction at the edges?
- Processing of graphene is rather tricky and contamination with substances, such as polymer or environmental substances, can be an issue. How do the authors proceed to get the clean interface between the graphene sheets?
- A graphical illustration of the experimental setup might be helpful for a reader, who is not in the field.

Reviewer #2 (Remarks to the Author):

By employing laser Raman microscopy upon stepwise loading the authors measured values of strain and estimated values of interlayer shear stress (ILSS) in the superlubricity regime over mm scale contact areas, under ambient conditions. The proposed experimental approach to study superlubricity is drastically different from previously used techniques, based on pulling a slider over a substrate and measuring forces. In addition, molecular dynamics (MD) simulations were performed to examine the influence of surface wrinkles on the stress transfer efficiency in bi-layer graphene, as well as to investigate a dependence of shear stress on the misfit angle between contacting layers and on the shear direction.

The effect of tensile stress on the Raman spectrum of graphene is well-known, and the relation allowing to estimate the shear stress from shear stain has been derived in the previous publication of the same authors. However, it should be emphasized that the proposed approach provides information only on the values of strain and stress averaged over surface area. In turn, it's well known (see for instance, van Wijk, et al., Phys. Rev. Lett. 113, 135504 (2014) and D. Mandelli, Sci.Rep. 7, 10851 (2017)) that due to the formation of moiré patterns at interfaces of incommensurate contacts the strain and stress distributions are highly nonuniform along the interface. And the measured friction force is determined by the highly stressed regions, rather than by the surface averaged value of the stress. Because of this I doubt that the proposed approach giving estimations only for the averaged stress may provide information on frictional properties of graphene bi-layer.

It's hard to understand how the MD simulations were done. The choice of interlayer potentials is crucial for correct description of friction (see for instance, the papers cited above), and it's not clear what potentials were used in this study. Did authors perform quasistatic or dynamic simulations? What contact sizes were considered in MD simulations? What boundary conditions were chosen?

Based on the above discussion, I believe that the presented results don't justify the publication of the manuscript in Nat. Comm.

Reviewer #3 (Remarks to the Author):

The paper should be published subject to the revisions outlined below. The experimental work is very good, but the interpretation is not always very clear.

Abstract (Lines 16-28). Please state that both exfoliated and CVD graphene have been studied. Lines 38-40. It is not clear if these mechanical properties are in response to in-plane stresses or transverse. Is the tensile strength of the graphite along a direction parallel or perpendicular to the layers? Is the compressive strength along a direction parallel or perpendicular to the layers? Lines 41-45. Are these frictional properties internal, between the graphene layers, or do they refer to the use of graphite or graphene as a lubricant?

Lines 46-63. There is no mention here of the role of water as a lubricant in graphite, bilayer graphene and monolayer graphene. This is a major omission that must be discussed. It is known that water from the atmosphere enables the lubrication properties of graphite, and that if the relative humidity is very low then graphite is no longer a lubricant. Atmospheric water adsorbs on the surface of graphene and also intercalates between the layers of bilayer graphene. There are recent publications on this.

Lines 22-28 (Abstract) and lines 58-60. Please clarify. The Abstract (lines 22-24) states that the random stacking, the presence of wrinkles, ... between two graphene layers are considered responsible for the facile shearing at the macroscale, yet lines 58-60 state that lubricant behavior has been observed for single layer graphene and even at the macroscale.

Lines 64 -67. Please state here if your bilayer graphene is made from exfoliated or CVD graphene, because in the Results section both are used. Exfoliated graphene is much more perfect than CVD graphene. The authors state that the single 2D Raman peak indicates a bilayer with AA stacking. But it might also indicate a defective monolayer, so that you have two monolayers (which don't interact) rather than a bilayer (two layers which do interact). Please discuss.

Line 71. Similarly, if one assigns a 2D peak to one of the graphene layers in a bilayer, then they become essentially two monolayers. If the stress could still be transferred to the top layer, then one could assign all the 2D components (4 peaks) to the bilayer, consistent with Bernal (AB) stacking. Please discuss.

Line 109. Whether the 2D line is symmetric or not depends on whether the two layers interact with each other (two monolayers or one bilayer).

Line 110. The 2D peak is fitted with two Lorentzians. However, it is usual to use 4 peaks to fit the 2D profile of bilayer graphene, which may give a better fit. I suggest the authors do this.

Figure 2. The frequency of the 2D peak of graphene is plotted against the strain of the substrate. However, slip between the graphene and the substrate is well known, both suddenly (stick-slip) and gradually. Please comment.

"Tunable macroscale structural superlubricity in two-layer graphene via strain engineering"

Submission to Nature Communications - NCOMMS-19-734928-T

CONTENTS

REPLY TO THE COMMENTS MADE BY THE REFEREES	2
REVISED PAPER: CHANGES AND ADDITIONS MADE TO ORIGINAL MANUSCRIPT.....	Error! Bookmark not defined.

REPLY TO THE COMMENTS MADE BY THE REFEREES

Reviewer #1:

Comment: “The paper describes a novel method to observe frictional properties between graphene sheets. The method is based upon Raman imaging, where the strain in the film leads to shifts of the Raman peaks. Since the graphene sheets are deposited on a flexible beam, the strain can be continuously changed. Interestingly, the authors observe that the top graphene layer shows only very small strain values, which means that the top layer starts to slide above a certain threshold. Calculations of the effective shear stress show low values of the order of 0.1 MPa, which are in agreement with other methods. Therefore, I propose to publish this interesting paper with minor changes.”

Reply: We thank the reviewer for the careful reading of our work and the recommendation for publication in Nature Communications. All the points raised have been fully addressed as given in detail below.

Comment: “The strain distribution shows maxima towards the edges. It would be interesting to explain this observation in more detail. Is it related to influence of the enhanced friction at the edges?”

Reply: The strain build-up from the edges towards the centre of the flakes is characteristic of the stress transfer mechanism through interfacial shearing. Basically a finite length is required for the efficient transfer of strain to graphene. In mechanics this is termed as shear-lag effect and is the prevailing mechanism of stress-transfer from matrix to inclusions in composite materials[1, 2]. For the case of graphene-polymer interface the strain transfer occurs through friction with constant interlayer shear stress[3]. This issue was recently explained in depth in our earlier paper[3]. This point has been further clarified by adding the following text:

“The shape of the stress-transfer curve from the polymer to the inclusion for the bottom layer (figure 3a) is, as expected, governed by polymer–graphene shearing, that leads to stress build up from the flake edges and the attainment of a plateau at the middle of the flake²³. This mechanism is a result of the strain transfer with friction, which leads to linear strain profiles at the edges with constant interlayer frictional stress and the length required for strain transfer increases with the increase in the applied load³¹. As it is discussed below this is a crucial point that has not received attention and holds for the case of a graphene-graphene interface.”

Comment: “Processing of graphene is rather tricky and contamination with substances, such as polymer or environmental substances, can be an issue. How do the authors proceed to get the clean interface between the graphene sheets?”

Reply: In order to get clean surfaces the CVD graphenes were rinsed in water for 4-5 times. This is a standard procedure when transferring CVD graphene from copper substrate. After transferring the second CVD on the polymer/graphene, the sample was dried for the removal of any water molecules. Regarding the exfoliated specimen, before the deposition of the tape with the graphitic materials the surface of the polymer was cleaned with IPA and dried with nitrogen. The freshly spin coated PMMA layers were attached together quickly and thus they were minimally exposed to the environment and no contamination is expected for such quick procedure. Moreover, the high shift rate of the 2D Raman peak shows that the interface was robust and no interfacial problems were encountered. The following text is added:

Revised text:

“The graphenes were rinsed with distilled water four to five times⁵⁹ in order to clean their surfaces. Extra caution was taken in order the top layer is supported only by the bottom graphene. This was succeeded by depositing a relatively large CVD to the bottom (i.e ~2 x 1 cm for length and width, respectively), and for the top layer about half the dimensions of the bottom layer. As mentioned below, the initial CVD sample has dimension of 7 cm x 7 cm square, thus no size limitations were encountered during this procedure. Finally the sample left to dry under nitrogen flow and zero relative humidity over twenty four hours for the removal of any water molecules. We note that we did not subject the sample to heat to avoid the potential compressive strain induced by heating.”

Comment: “A graphical illustration of the experimental setup might be helpful for a reader, who is not in the field.”

Reply: A new schematic has been added to the draft, illustrating in detail the experimental setup. The schematic is added to the figure 1 in the main text. In the schematic the graphene/graphene interface is of incommensurate stacking and the Moire patterns are distinguished. The following schematic is now figure 1d in the main text:

The following text is added:

“In figure 1d a schematic of the experimental setup is shown with two single layer graphenes stacked in an incommensurate state. The figure depicts also the formation of Moiré patterns as discussed in detail below.”

Reviewer #2:

Comment: “By employing laser Raman microscopy upon stepwise loading the authors measured values of strain and estimated values of interlayer shear stress (ILSS) in the superlubricity regime over mm scale contact areas, under ambient conditions. The proposed experimental approach to study superlubricity is drastically different from previously used techniques, based on pulling a slider over a substrate and measuring forces. In addition, molecular dynamics (MD) simulations were performed to examine the influence of surface wrinkles on the stress transfer efficiency in bi-layer graphene, as well as to investigate a dependence of shear stress on the misfit angle between contacting layers and on the shear direction.”

Reply: We thank the reviewer for the careful reading of our work and the brief account of our approach given above.

Comment: “The effect of tensile stress on the Raman spectrum of graphene is well-known, and the relation allowing to estimate the shear stress from shear stain has been derived in the previous publication of the same authors. However, it should be emphasized that the proposed approach provides information only on the values of strain and stress averaged over surface area. In turn, it's well known (see for instance, van Wijk, et al., Phys. Rev. Lett. 113, 135504 (2014) and D. Mandelli, Sci.Rep. 7, 10851 (2017)) that due to the formation of moiré patterns at interfaces of incommensurate contacts the strain and stress distributions are highly nonuniform along the interface. And the measured friction force is determined by the highly stressed regions, rather than by the surface averaged value of the stress. Because of this I doubt that the proposed approach giving estimations only for the averaged stress may provide information on frictional properties of graphene bi-layer. Based on the above discussion, I believe that the presented results don't justify the publication of the manuscript in Nat. Comm.”

Reply: In mechanics the definition of a friction force which is the main thesis of our paper is certainly related to the surface area that is been loaded under shear. The reviewer seems to compare our findings with possible stress concentration points observed in a heterogeneous system (Gr/ hBN) by atomistic simulations (e.g. paper by van Wijk et al). We believe that the comparison between our Gr/ Gr system with simulated heterostructures is inappropriate and the conclusion that our experimental work which is also backed up by computer simulations should not be published is not at all substantiated. The main conclusion drawn in the PRL paper that the reviewer mentions, is that the Moiré patterns and the strain distribution in the graphene-hBN heterojunction depend on the interactions between the carbon-nitrogen and carbon-boron atoms, thus we do not see how this paper is related to our work! Instead as argued below we have included in the revised version the results of the simulations as

above and explain the differences in terms of scale of measurement and system characteristics between our work and the papers above.

Friction itself at the scale of our study is indeed an average process that governs the superlubric behaviour of our Gr/ Gr system. If there was an area with highly concentrated stress or strain, then this point would act as a pin (and consequently would result in interlocking and no facile shearing) and would drag along the other graphene layer. In fact, this is acknowledged in the paper by van Wijk et al., who claim that for incommensurate stacking“... *the corresponding heterogeneous junction exhibits a transition from stick-slip motion to smooth sliding as a function of contact size.*” In other words, the points of high concentration if present would act as singularities leading to smooth sliding as observed in our work. These simulation results also agree with a recent *Nature Materials* paper [5] which is concerned with hBN/graphene interfaces. It was observed experimentally that even the aligned contact (i.e for lock-in shearing direction), which corresponds to pin effects and stress concentration, also presents superlubric behaviour due to the inherent lattice constant mismatch, with the only difference being the observed higher interlayer frictional shear stress (however still in the superlubric regime [5]). In our case, the incommensurability is induced by the tensile strain. A more recent theoretical study on the same system as ours [6]shows that 0.1% of strain reduces the interlayer friction by two orders of magnitude because of the Moiré patterns and moreover, the interlayer shear forces in the strained graphene/graphene interface are reduced with the increase in graphene size, further supporting our findings!

In the revised version we have added the observations between our work and the papers mentioned by the reviewer and the relative papers as follows:

“We note here that two lattices in incommensurate state form Moiré patterns that affect the spatial distribution of strain and consequently the interlayer shearing depends on the Moiré characteristics⁴¹⁻⁴³. Despite the stress concentration that might be present in such cases, the mismatch in the lattice constant between the two layers eliminates such effects. For example, as explained in the case of hBN/graphene interface with inherent lattice constant mismatch¹⁸, the possible presence of small friction anisotropy does not alter the overall system behaviour. Moreover, MD simulations on shearing a graphene layer on a strained graphene, show that the friction dramatically decreases with the increase in graphene size. This is because the large contact areas result in a much larger length than that of Moiré patterns and the friction force tends to the value of the incommensurate state³⁹. Thus, such effects can hardly affect the interlayer shear stress measured at the micron/ mm scale of our experiments.

Finally, we have added a new schematic to the draft, illustrating in detail the experimental setup. The schematic is added to the figure 1 in the main text. In the schematic the graphene/graphene interface is of incommensurate stacking and the

Moiré patterns are distinguished. The following schematic is now figure 1d in the main text:

The following text is added:

“In figure 1d a schematic of the experimental setup is shown with two single layer graphenes stacked in an incommensurate state. The figure depicts also the formation of Moiré patterns as discussed in detail below.”

Comment: “It's hard to understand how the MD simulations were done. The choice of interlayer potentials is crucial for correct description of friction (see for instance, the papers cited above), and it's not clear what potentials were used in this study. Did authors perform quasistatic or dynamic simulations? What contact sizes were considered in MD simulations? What boundary conditions were chosen?”

Reply: We thank the reviewer for raising this issue as some details of the simulations had not been presented in the submitted manuscript and supporting information. We now include a full reference to the LCBOP potential that was used, along with meticulous technical details on the simulations' setups and execution. We seized this opportunity and have repeated the simulations employing the AIREBO potential with exactly the same setups and procedures as the initial simulations. Initial sample sizes were adjusted to accommodate that the AIREBO potential underestimates the lattice constant of graphene. We once again find a significant decrease in ISS as the top layer passes over the wrinkled regions of the substrate, as well as in the chiral simulations for orientations other than the 0° case. This is in itself noteworthy since the two potentials have significantly different formulations.

Regarding the term "quasi-static" used by the reviewer this, in our opinion, can lead to significant misinterpretation. As clearly stated, we employed MD simulations in which the top sheet is displaced at a very low velocity (or course, with a significant increase of the computational cost and simulation runtime), and is not pulled by an applied force. Nevertheless, we recognize the point made by the Reviewer that one may be erroneously led to interpret the usage of low velocity as a type of quasi-static computation, in the sense that it may be perceived that a slowly varying quantity seemingly exists. Also, there is no artificial 'dead-time' to allow the structure to relax

between displacements, and no algorithmic interventions in the integrations have been used, for example interpolations or local optimizations (such as FIRE) between quasi-static states. We have added the following references with respect to this:

Mandelli, D., Leven, I., Hod, O. & Urbakh, M. Sliding friction of graphene/hexagonal-boron nitride heterojunctions: a route to robust superlubricity. Scientific reports 7, 10851 (2017).

Van Wijk, M., Schuring, A., Katsnelson, M. & Fasolino, A. Moiré patterns as a probe of interplanar interactions for graphene on h-BN. Physical review letters 113, 135504 (2014).

All the aforementioned are implicit in our reference of the simulations as being MD that are further clarified by the so provided setup details. As such, there should be no need to clarify as them not being "quasi-static". However, to avert any possibility of misunderstanding, in line with the Reviewer's observation, we now explicitly clarify this point in the Simulations section. Also, we now provide additional information on details of the MD setup, that will further avert any misunderstanding, and should fully satisfy the Reviewer's query.

As already mentioned, we now provide meticulous technical details on the simulations' setups and execution. This includes tabulated dimensions of the top sheets in all cases, from which the nominal contact areas are readily derived. In the initial version of the manuscript this information was only provided for the sliding simulation over the wrinkled layer. Periodic boundary conditions were used in all cases, throughout. The bottom layer is periodic in both directions, and the top layer is periodic in the direction normal to its displacement over the corrugated rigid bottom layer.

On this basis we have added the new results in the SI and a brief discussion in the main text. The full details of all our calculations have been added. In the main text the following is added:

The effect is also captured by repeating the simulations employing the AIREBO potential³³, as discussed in the Supporting Information^{18,54-57}.

Molecular dynamics simulations were performed employing the LCBOP⁶⁰ potential that offers Morse type long-range interactions that exclude nearest neighbours and offers suitably parametrized short-range term, that do not lead to unrealistic structural defects. The simulations were fully dynamic for the dynamic particles of the system (as opposed to quasi-static simulations that employ additional algorithmic relaxation schemes^{42,43}. Periodic boundary conditions were used in all cases, throughout. The bottom layer is periodic in both directions was corrugated through biaxial compression and remained rigid during the sliding stage, as detailed in the Supporting Information. The top layer was periodic in the direction normal to its displacement over the corrugated rigid bottom layer. The components of the forces along the displacement

direction acting on all of the top-layer atoms were summed and averaged every 2000 time steps.

For the “chiral” set of simulations computational cells of different sizes were used (provided in the Supporting Information), each adapted to conform to the sheet size constraints imposed by a given chiral angle.

Computational cells in the Supporting Information were visualized using OVITO[10].

Figure 6 was redrawn.

And in the Supporting Information the following sections have been added/updated:

To illustrate the effect of wrinkles on the interlayer shearing between two graphene layers, molecular dynamics simulations were performed, using the LCBOP potential¹ on supported bilayer graphene (as is the actual sample) that had been previously subjected to biaxial compressive strain of -0.6% in order to induce the formation of wrinkles. The compression was induced by reducing the computation cell (box) at a constant engineering strain rate of $-0.005\%/ps$ of its original dimensions, realized every 100fs. During the compressive stage the bottom layer was also at a temperature of 300K so thermal ripples were also imprinted in the overall corrugation, and kept rigid thereafter (for a comprehensive exposition on the validity of the rigid-substrate approach we refer to Ref. 2).

The displacement of the top layer was performed with a small velocity of $0.02 \text{ \AA}/ps$ (which is in the range of the lowest used in the literature) and realized through the right-most edge (as shown in figure S6a).

3.2 Alternative potential

*The effect is also captured when the simulations are repeated employing the AIREBO potential⁴, despite the significant differences in the formulation of the two potentials. The extent of the long-range Lennard–Jones interaction for the AIREBO potential were set to 3σ , which translates to 10.2\AA . The AIREBO potential underestimates the lattice constant of graphene compared to LCBOP (and experiment), that leads to smaller areas of the graphenes by a factor of ~ 0.9688 at a temperature of 300K. Due to the much higher computational cost compared to the LCBOP potential, the simulation time was such to clearly demonstrate the ISS reduction when passing over the formed wrinkle perpendicular to the displacement direction. The graph of **figure S6** reveals significant ILSS drops as the top layer passes over the wrinkled region of the bottom layer.*

Figure S6. (left) *Wrinkled bilayer graphene. Three replications of the computational cell are shown at the initial state prior to shifting. The top layer is shifted along the y-axis (left to right as shown in figure). The dimensions of the bilayer sheet are 39.3 nm x 39.1 nm and has a total of 75978 atoms. The top layer has one fourth the overall length along the y-axis. Simulations were performed using the AIREBO potential. (right) Evolution of the ILSS with respect to the displacement of the top layer graphene.*

As in the case of the LCBOP potential, the reduction in ILSS is also noted when using the AIREBO potential for chiral top layers. The results are shown in figure S7. The dimensions of the sheets that correspond to the LCBOP potential are provided in the table S1. In the case of the AIREBO potential each dimension is scaled by a factor of 0.9843 due to the underestimation of the lattice constant.

Table S1. *Dimensions of the top layer used in each of the ‘chiral’ simulations when employing the LCBOP potential, number of atoms of the top layer, and overall number of atoms of the simulation cell.*

Chiral Angle	L_x (nm)	L_y (nm)	Number of atoms Top Layer	Number of atoms Simulation Cell
AC	20.136	4.537	3608	19024
ZZ	7.160	4.543	1292	3960
7.589°	19.239	3.675	2736	9912
15.295°	14.488	4.182	2328	7756
23.413°	11.296	2.767	1204	5436

Figure S7. ILSS values for various chiral directions at a temperature of 300 K. It is observed that when shearing a mono-layer graphene in achiral directions relative to the underline graphene, the stick-slip motion breaks down and the ILSS is significantly decreased. Performed employing the AIREBO potential.

3.3 Thermostat dampening.

The employed model and the usage of large cells produces clear results that are not masked by thermal fluctuations, as shown in **figure 6** of the main text and **figures S5–S7**. The temperature is controlled by means of a Nosé–Hoover thermostat with a

temperature dampening parameter of $T_{damp}=0.1ps$. With this coupling strength the temperature is stabilized efficiently, as shown in **figure S8**. To obtain some insight on possible dampening of the dynamics in connection with the temperature dampening strength^{2,5-8}, simulations were performed starting with a dampening parameter of $T_{damp}=0.1ps$ which at some point during the simulation was switched to a larger value. Specifically, in **figure S9a** the simulation starts with $T_{damp}=0.1ps$ and after 500ps (one million time steps) switches to $T_{damp}=1.0 ps$, and the recorded ISS values remained unaffected. The same was observed for larger value of $T_{damp}=4.0ps$ (**figures S9b**). Usage of larger damping factors within this approach leads to inefficient energy dissipation.

Figure S8. Instantaneous temperatures the simulation of the top layer sliding over the bottom layer, using the (a) LCBOP and (b) AIREBO potential. The Nosé–Hoover thermostat dampening parameter was set to $T_{damp}=0.1ps$.

Figure S9. Evolution of the ILSS with respect to the displacement of the top layer graphene. The Nosé–Hoover thermostat dampening parameter was switched from $T_{damp}=0.1ps$ to (a) $T_{damp}=1.0 ps$, (b) $T_{damp}=4.0 ps$, after a displacement of 10 Å (i.e., after 500ps). Simulations were performed using the LCBOP potential.

In addition to a proper reference to the LCBOP potential, we now include the following references:

Mandelli, D., Leven, I., Hod, O. & Urbakh, M. *Sliding friction of graphene/hexagonal-boron nitride heterojunctions: a route to robust superlubricity. Scientific reports* **7**, 10851 (2017).

Van Wijk, M., Schuring, A., Katsnelson, M. & Fasolino, A. *Moiré patterns as a probe of interplanar interactions for graphene on h-BN. Physical review letters* **113**, 135504 (2014).

Los, J.; Fasolino, A., *Intrinsic long-range bond-order potential for carbon: Performance in Monte Carlo simulations of graphitization. Physical Review B* **2003**, 68 (2), 024107.

Stuart, S. J.; Tutein, A. B.; Harrison, J. A., *A reactive potential for hydrocarbons with intermolecular interactions. The Journal of chemical physics* **2000**, 112 (14), 6472-6486.

Vanossi, A.; Manini, N.; Urbakh, M.; Zapperi, S.; Tosatti, E., *Colloquium: Modeling friction: From nanoscale to mesoscale. Reviews of Modern Physics* **2013**, 85 (2), 529-552.

Vanossi, A.; Manini, N.; Urbakh, M.; Zapperi, S.; Tosatti, E., *Colloquium: Modeling friction: From nanoscale to mesoscale. Reviews of Modern Physics* **2013**, 85 (2), 529-552.

Basconi, J. E.; Shirts, M. R., *Effects of Temperature Control Algorithms on Transport Properties and Kinetics in Molecular Dynamics Simulations. Journal of Chemical Theory and Computation* **2013**, 9 (7), 2887-2899.

Smith, E. D.; Robbins, M. O.; Cieplak, M., *Friction on adsorbed monolayers. Physical Review B* **1996**, 54 (11), 8252-8260.

Tomassone, M. S.; Sokoloff, J. B.; Widom, A.; Krim, J., *Dominance of Phonon Friction for a Xenon Film on a Silver (111) Surface. Physical Review Letters* **1997**, 79 (24), 4798-4801.

Stukowski, A. *Visualization and analysis of atomistic simulation data with OVITO—the Open Visualization Tool. Modelling and Simulation in Materials Science and Engineering* **18**, 015012, doi:10.1088/0965-0393/18/1/015012 (2009).

Reviewer #3:

Comment: “The paper should be published subject to the revisions outlined below. The experimental work is very good, but the interpretation is not always very clear.”

Reply: We thank the reviewer for his careful and extensive reading, as well his insightful suggested improvements of our work and the clear recommendation for publication in Nature Communications. All the points raised have been fully addressed as given in detail below.

Comment: “Abstract (Lines 16-28). Please state that both exfoliated and CVD graphene have been studied”.

Reply: The Abstract has been amended and now reads as follows:

*“Here, in contrast, we show the presence of macro-scale structural superlubricity between two randomly stacked graphene layers **produced by both mechanical exfoliation and CVD** upon the imposition of a tensile stress.”*

Comment: “Lines 38-40. It is not clear if these mechanical properties are in response to in-plane stresses or transverse. Is the tensile strength of the graphite along a direction parallel or perpendicular to the layers? Is the compressive strength along a direction parallel or perpendicular to the layers? Lines 41-45. Are these frictional properties internal, between the graphene layers, or do they refer to the use of graphite or graphene as a lubricant?”

Reply: Lines 38-40. We refer to axial (in plane) mechanical properties for both tensile and compressive loadings. Lines 41-45. These are internal frictional properties between adjacent graphene layers. The revised text now reads:

*“From the mechanics point of view, the **in-plane** tensile fracture strength tends to decrease with the increase in thickness and recent experiments have also shown a decrease in the **in-plane** compressive strength as a result of premature cohesive shear failure.”*

*“Experiments performed using Friction Force Microscopy (FFM) **by shearing an AFM tip over the surface of 2D crystals (graphene, hBN etc.) with various numbers of layers in thickness** have indicated that 2D materials possess thickness dependent friction properties, and for graphene the friction has been found to increase with the decrease in thickness⁴. Due to the significance for **the use of graphitic materials as lubricants in a number of applications, the friction behaviour of graphene^{5, 6, 7, 8, 9, 10, 11} and graphite^{12, 13, 14, 15}** has been the subject of extensive research.”*

Comment: Lines 46-63. There is no mention here of the role of water as a lubricant in graphite, bilayer graphene and monolayer graphene. This is a major omission that must be discussed. It is known that water from the atmosphere enables the lubrication properties of

graphite, and that if the relative humidity is very low then graphite is no longer a lubricant. Atmospheric water adsorbs on the surface of graphene and also intercalates between the layers of bilayer graphene. There are recent publications on this.

Reply: We thank the reviewer for this insightful comment. We mention this point now in the introduction:

“Another issue that plays a crucial role in the lubricant properties of graphene and graphite is the presence of water. Recent studies show that high relative humidity enhances the lubricant behaviour of graphene on SiO₂²¹, and also water can be intercalated between two graphene layers and affect the interlayer interactions²².”

Moreover, the presence of humidity might also explain some differences between experiment and MD simulations. The following text is added:

“Another factor that could affect the interlayer interactions is the presence of relative humidity, which could be a source for the observed discrepancies regarding graphene superlubricity between simulations and experiments²².”

We also clarify another point regarding the preparation of the samples to avoid trapped water in the interfaces. The following text is added:

“The graphenes were rinsed with distilled water four to five times⁵⁹ to clean their surfaces. Extra caution was taken in order the top layer is supported only by the bottom graphene. This was followed by the deposition of a relatively large CVD to the bottom (i.e ~2 x 1 cm for length and width, respectively), and for the top layer about half the dimensions of the bottom layer. As mentioned below, the initial CVD sample has dimension of 7 cm x 7 cm square, thus no size limitations were encountered during this procedure. Finally the sample left to dry under nitrogen flow and with zero relative humidity over twenty four hours for the removal of any water molecules. We note that we did not subject the sample to heat to avoid the potential compressive strain induced by heating.”

Comment: “Lines 22-28 (Abstract) and lines 58-60. Please clarify. The Abstract (lines 22-24) states that the random stacking, the presence of wrinkles, ... between two graphene layers are considered responsible for the facile shearing at the macroscale, yet lines 58-60 state that lubricant behavior has been observed for single layer graphene and even at the macroscale.”

Reply: We thank the reviewer for his/her observation. This comment refers to macro-scale superlubricity of graphene when it is used along with nanodiamond particles and diamond like carbon coating[13]. We have cited the correct article now and clarified this point in the revised text as follows:

“Lubricant behaviour has also been observed for single layer graphene and even at the macro-scale when sliding a surface coated with graphene against another surface with diamond like carbon and nanodiamond particles¹⁹, which makes graphene a very versatile, thin and transparent coating material for use in a variety of applications.”

Comment: “Lines 64 -67. Please state here if your bilayer graphene is made from exfoliated or CVD graphene, because in the Results section both are used. Exfoliated graphene is much more perfect than CVD graphene. The authors state that the single 2D Raman peak indicates a bilayer with AA stacking. But it might also indicate a defective monolayer, so that you have two monolayers (which don't interact) rather than a bilayer (two layers which do interact). Please discuss.”

Reply: Both exfoliated and CVD graphene bilayers were examined. This is now explicitly stated as follows:

“Herein, we report direct measurements of the interlayer shear stress in an incommensurately stacked bi-layer graphene produced both by mechanical exfoliation of graphite and chemical vapour deposition (CVD) synthesis simply supported on a polymer substrate.”

The fact that the two graphene layers interact can be deduced from the changes in the 2D Raman peak. The enhancement in the intensity and the changes in the FWHM and position strongly suggest that the two graphene layers interact. Moreover, the most convincing argument that the two graphene layers do interact, is the strain transferred that is clearly observed by the Raman shifts; if the opposite was true there would be no strain transfer between the graphene layers. We have clarified this point by adding the following text:

“The changes observed in the line-shape and frequency of the 2D peak, as well the strain transferred which is clearly demonstrated through the Raman shift show that the two graphene layers are in contact with each other.”

Comment: “Line 71. Similarly, if one assigns a 2D peak to one of the graphene layers in a bilayer, then they become essentially two monolayers. If the stress could still be transferred to the top layer, then one could assign all the 2D components (4 peaks) to the bilayer, consistent with Bernal (AB) stacking. Please discuss. Line 109. Whether the 2D line is symmetric or not depends on whether the two layers interact with each other (two monolayers or one bilayer). Line 110. The 2D peak is fitted with two Lorentzians. However, it is usual to use 4 peaks to fit the 2D profile of bilayer graphene, which may give a better fit. I suggest the authors do this.”

Reply: We agree with the reviewer that a bi-layer of Bernal stacking is fitted with four Lorentzian curves. In our case, the stacking is random and we observe a single peak similar to the single layer. The full width at half maximum (FWHM) for the two-layer

with random stacking is smaller than the FWHM of the single layer, and we also observe an enhancement of the 2D peak intensity. These phenomena have been examined in other studies for folded and scrolled graphene[14, 15]. Moreover, we have attempted fits with 4 Lorentzian curves which have shown that the sub-peaks tend to merge thus no additional information that can be extracted from such analysis. We further discuss and analyse this point in the supporting information and we also present 2D peak spectra from a Bernal stacked bilayer for a direct comparison.

Comment: “Figure 2. The frequency of the 2D peak of graphene is plotted against the strain of the substrate. However, slip between the graphene and the substrate is well known, both suddenly (stick-slip) and gradually. Please comment.”

Reply: In the case presented herein, the slip is sudden as evident by the abrupt drop of the 2D peak frequency. We have clarified this point and made the following additions:

“The position of the 2D peak drops to lower wavenumbers abruptly from one strain level to the next, suggesting that slipping between the graphene and the polymer occurs after this point.”

Point-by-point changes in the draft:

Abstract (added text): Here, in contrast, we show the presence of macro-scale structural superlubricity between two randomly stacked graphene layers **produced by both mechanical exfoliation and CVD** upon the imposition of a tensile stress.

Page 2 (Added text): From the mechanics point of view, the **in-plane** tensile fracture strength tends to decrease with the increase in thickness and recent experiments have also shown a decrease in the **in-plane** compressive strength as a result of premature cohesive shear failure.

Page 2 (Added text): Experiments performed using Friction Force Microscopy (FFM) **by shearing an AFM tip over the surface of 2D crystals (graphene, hBN etc.) with various numbers of layers in thickness** have indicated that 2D materials possess thickness dependent friction properties, and for graphene the friction has been found to increase with the decrease in thickness⁴. Due to the significance for **the use of graphitic materials as lubricants** in a number of applications, the friction behaviour of graphene^{5, 6, 7, 8, 9, 10, 11} and graphite^{12, 13, 14, 15} has been the subject of extensive research.

Page 3 (Added text): Lubricant behaviour has also been observed for single layer graphene and even at the macro-scale **when sliding a surface coated with graphene against another surface with diamond like carbon and nanodiamond particles**¹⁹, which makes graphene a very versatile, thin and transparent coating material for use in a variety of applications.”

Page 3 (Added text): Herein, we report direct measurements of the interlayer shear stress in an incommensurately stacked bi-layer graphene produced both by mechanical exfoliation of graphite and chemical vapour deposition (CVD) synthesis simply supported on a polymer substrate.”

Page 3 (Added text): *Another issue that plays a crucial role in the lubricant properties of graphene and graphite is the presence of water. Recent studies show that high relative humidity enhances the lubricant behaviour of graphene on SiO₂²¹, and also water can be intercalated between two graphene layers and affect the interlayer interactions²².*

Page 5-6: (Added figure and added text in figure caption): In figure 1d a schematic of the experimental setup is demonstrated with two single layer graphenes stacked in an incommensurate state. The figure depicts also the formation of Moiré patterns as discussed in detail below.

Page 7 (Added text): The changes observed in the line-shape and frequency of the 2D peak, as well the strain transferred which is clearly demonstrated through the Raman shift show that the two graphene layers are in contact with each other.”

Page 7 (Added text): The shape of the stress-transfer curve from the polymer to the inclusion for the bottom layer (figure 3a) is, as expected, governed by polymer–graphene shearing, that leads to stress build up from the flake edges and the attainment of a plateau at the middle of the flake²³. This mechanism is a result of the strain transfer with friction, which leads to linear strain profiles at the edges with constant interlayer frictional stress and the length required for strain transfer increases with the increase in the applied load³¹. As it is discussed below this is a crucial point that has not received attention and holds for the case of a graphene-graphene interface.”

Page 7 (Added text): The position of the 2D peak drops to lower wavenumbers abruptly from one strain level to the next, suggesting that slipping between the graphene and the polymer occurs after this point.”

Page 14 (Added text): We note here that two lattices in incommensurate state form Moiré patterns that affect the spatial distribution of strain and consequently the interlayer shearing depends on the Moiré characteristics⁴¹⁻⁴³. Despite the stress concentration that might be present in such cases, the mismatch in the lattice constant between the two layers eliminates such effects. For example, as explained in the case of hBN/graphene interface with inherent lattice constant mismatch¹⁸, the possible presence of small friction anisotropy does not alter the overall system behaviour. Moreover, MD simulations on shearing a graphene layer on a strained graphene, show that the friction dramatically decreases with the increase in graphene size. This is because the large contact areas result in a much larger length than that of Moiré patterns and the friction force tends to the value of the incommensurate state³⁹. Thus, such

effects can hardly affect the interlayer shear stress measured at the micron/ mm scale of our experiments.

Figure 6 was redrawn.

Page 21 (added text): Another factor that could affect the interlayer interactions is the presence of relative humidity, which could be a source for the observed discrepancies regarding graphene superlubricity between simulations and experiments²².

Page 22 (added text): “The graphenes were rinsed with distilled water four to five times⁵⁹ in order to clean their surfaces. Extra caution was taken in order the top layer is supported only by the bottom graphene. This was succeeded by depositing a relatively large CVD to the bottom (i.e ~2 x 1 cm for length and width, respectively), and for the top layer about half the dimensions of the bottom layer. As mentioned below, the initial CVD sample has dimension of 7 cm x 7 cm square, thus no size limitations were encountered during this procedure. Finally the sample left to dry under nitrogen flow and zero relative humidity over twenty four hours for the removal of any water molecules. We note that we did not subject the sample to heat to avoid the potential compressive strain induced by heating.”

Page 24 added: The effect is also captured by repeating the simulations employing the AIREBO potential⁵³, as discussed in the Supporting Information^{18,54-57}.

The MD simulations were performed employing the LCBOP⁶⁰ potential that offers Morse type long-range interactions that exclude nearest neighbours and offers suitably parametrized short-range term, that do not lead to unrealistic structural defects. The simulations were fully dynamic for the dynamic particles of the system (as opposed to quasi-static simulations that employ additional algorithmic relaxation schemes^{42,43}). Periodic boundary conditions were used in all cases, throughout. The bottom layer is periodic in both directions was corrugated through biaxial compression and remained rigid during the sliding stage, as detailed in the Supporting Information. The top layer was periodic in the direction normal to its displacement over the corrugated rigid bottom layer. The components of the forces along the displacement direction acting on all of the top-layer atoms were summed and averaged every 2000 time steps.

For the “chiral” set of simulations computational cells of different sizes were used (provided in the Supporting Information), each adapted to conform to the sheet size constraints imposed by a given chiral angle.

Computational cells in the Supporting Information were visualized using OVITO⁶².

Page 25 Author’s contribution added: CA, GP and GT prepared and characterized the samples and CA performed the tensile experiments. ENK performed the MD simulations. CG supervised the project. CA, ENK and CG wrote the paper with input from all authors.

Page 25 Data availability added:

The data that support the findings of the present study are available within the paper and its Supplementary Information file. Other data are available from the corresponding authors upon request.

- [1] H. Cox, British journal of applied physics, 3 (1952) 72.
- [2] G. Anagnostopoulos, C. Androulidakis, E.N. Koukaras, G. Tsoukleri, I. Polyzos, J. Parthenios, K. Papagelis, C. Galiotis, ACS applied materials & interfaces, 7 (2015) 4216-4223.
- [10] A. Stukowski, Modelling and Simulation in Materials Science and Engineering, 18 (2009) 015012.
- [14] Z. Ni, Y. Wang, T. Yu, Y. You, Z. Shen, Physical Review B, 77 (2008) 235403.
- [15] R. Podila, R. Rao, R. Tsuchikawa, M. Ishigami, A.M. Rao, Acs Nano, 6 (2012) 5784-5790.
- [18] Y. Song, D. Mandelli, O. Hod, M. Urbakh, M. Ma, Q. Zheng, Nature materials, 17 (2018) 894.
- [19] D. Berman, S.A. Deshmukh, S.K. Sankaranarayanan, A. Erdemir, A.V. Sumant, Science, 348 (2015) 1118-1122.
- [21] T. Arif, G. Colas, T. Filletter, ACS applied materials & interfaces, 10 (2018) 22537-22544.
- [22] A. Qadir, Y. Sun, W. Liu, P.G. Oppenheimer, Y. Xu, C. Humphreys, D. Dunstan, Physical Review B, 99 (2019) 045402.
- [31] C. Androulidakis, D. Surlantzis, E. Koukaras, A. Manikas, C. Galiotis, Nanoscale Advances, 1 (2019) 4972-4980.
- [39] K. Wang, C. Qu, J. Wang, W. Ouyang, M. Ma, Q. Zheng, ACS applied materials & interfaces, 11 (2019) 36169-36176.
- [41] C. Woods, L. Britnell, A. Eckmann, R. Ma, J. Lu, H. Guo, X. Lin, G. Yu, Y. Cao, R. Gorbachev, Nature physics, 10 (2014) 451.
- [42] D. Mandelli, I. Leven, O. Hod, M. Urbakh, Scientific reports, 7 (2017) 10851.
- [43] M. Van Wijk, A. Schuring, M. Katsnelson, A. Fasolino, Physical review letters, 113 (2014) 135504.
- [59] J.W. Suk, A. Kitt, C.W. Magnuson, Y. Hao, S. Ahmed, J. An, A.K. Swan, B.B. Goldberg, R.S. Ruoff, ACS nano, 5 (2011) 6916-6924.